# Bacterial Co-Infection in Patients with COVID-19 Hospitalized (ICU and Not ICU): Review and Meta-Analysis

**DOI:** 10.3390/antibiotics11070894

**Published:** 2022-07-04

**Authors:** Adailton P. Santos, Lucas C. Gonçalves, Ana C. C. Oliveira, Pedro H. P. Queiroz, Célia R. M. Ito, Mônica O. Santos, Lilian C. Carneiro

**Affiliations:** 1Medicine College, Federal University of Goiás, 235 Street, Goiânia 74690-900, Brazil; apsantos1906@yahoo.com.br (A.P.S.); lucascandidogoncalves46@gmail.com (L.C.G.); chagaschagas@discente.ufg.br (A.C.C.O.); pedrodequeiroz@gmail.com (P.H.P.Q.); mosbio21@gmail.com (M.O.S.); 2Institute of Tropical Pathology and Public Health, Federal University of Goiás, 235 Street, Goiânia 74605-050, Brazil; crmalveste@gmail.com

**Keywords:** bacterial co-infections, COVID-19, SARS-CoV-2, public health

## Abstract

The prevalence of patients hospitalized in ICUs with COVID-19 and co-infected by pathogenic bacteria is relevant in this study, considering the integrality of treatment. This systematic review assesses the prevalence of co-infection in patients admitted to ICUs with SARS-CoV-2 infection, using the PRISMA guidelines. We examined the results of the PubMed, Embase, and SciELO databases, searching for published English literature from December 2019 to December 2021. A total of 542 rec ords were identified, but only 38 were eligible and, and of these only 10 were included. The tabulated studies represented a sample group of 1394 co-infected patients. In total, 35%/138 of the patients were co-infected with *Enterobacter* spp., 27% (17/63) were co-infected with methicillin-sensitive *Staphylococ cus aureus*, 21% (84/404) were co-infected with *Klebsiella* spp., 16% (47/678) of patients were co-infected with coagulase-negative *Staphylococcus*, 13% (10/80) co-infected with *Escherichia coli* (ESBL), and 3% (30/1030) of patients were co-infected with *Pseudomonas aeruginosa*. The most common co-infections were related to blood flow; although in the urinary and respiratory tracts of patients *Streptococcus pneumoniae* was found in 57% (12/21) of patients, coagulase negative *Staphylococcus* in 44% (7/16) of patients, and *Escherichia coli* was found in 37% (11/29) of patients. The present research demonstrated that co-infections caused by bacteria in patients with COVID-19 are a concern.

## 1. Introduction

Respiratory viral and bacterial infections contribute substantially to the global burden of morbidity and mortality. Such simultaneous infections with the flu virus or bacteria that cause pneumonia, tend to make the patient’s condition critical [1,2]. Although, critically ill patients rapidly develop acute respiratory distress syndrome and sepsis, leading to death from multiple organ failure. The main symptoms of COVID-19 are fever, fatigue, and dry cough. However, most patients have a good prognosis [3,4,5,6].

Bacterial co-infections associated with other coronaviruses, such as SARS-CoV-1 and MERS-CoV, have been reported in association with pandemic viruses at rates of 20–30%, respectively [2,7]. Bacterial co-infection is directly linked to increased morbidity and mortality from viral respiratory infections. Hospital admissions increase the risk of healthcare-associated infections (HCAI) which makes the disease more aggressive and difficult to treat, as well as inducing life-threatening complications and increasing the consumption of antibiotics [8,9].

Super infections and co-infections are commonly found in many respiratory diseases; viral infectious diseases and bacterial co-infections may be the cause of the increased mortality rate in patients infected with any viral infection [10,11,12,13]. Co-infection associated with viral pneumonia is the main cause of mortality and can considerably inhibit the host’s immune system, which decreases the pharmacological response and makes the prognosis of the disease harmful [14,15]. SARS-CoV-2 is a newly emerged pathogen that causes pneumonia with the possibility of worsening to hypoxic-type respiratory failure, organ failure, and acute kidney injury followed by myocarditis and thromboembolism. SARS-Cov-2 (COVID-19) leaves the body vulnerable to bacterial infections; however, this co-infection mechanism is not well understood but represents a threat to the respiratory epithelium favoring bacteremia [16,17,18,19,20,21]. A study carried out with ICU (Intensive Care Unit) patients in 88 countries showed that those patients who received at least one antibiotic during acute hospitalization, of these, more than half developed a secondary bacterial infection, requiring antibiotic therapy [22]. In China, 95% of patients and in the United Kingdom 80% of patients received antibiotics [23].

Antimicrobial resistance is seen as a major threat to public health, as well as to the economy and health security at the local and international levels. It is estimated that due to its spread across countries and continents the bacterial resistance increase will cause 10 million deaths annually by the year 2050 [24,25]. Relevant advances have been achieved and determined by the national AMR programme which is guided by the WHO Global Laboratory AMR Surveillance System (GLASS) in Uganda. Using the WHONET software [26], ARM data management was installed at the surveillance sites with trained personnel to guarantee the quality of the data. Six major pathogens that cause resistance-related deaths (*Escherichia*
*coli*, followed by *Staphylococcus aureus*, *Klebsiella pneumoniae*, *Streptococcus pneumoniae*, *Acinetobacter baumannii*, and *Pseudomonas aeruginosa*) were responsible for 929,000 deaths from ADR and 3.57 million (2.62–4.78) ADR-related deaths in 2019 [27,28]. Secondary infections predominantly involve a specific group of bacterial pathogens such as *S. aureus*, *Staphylococcus pneumoniae*, *Streptococcus pyogenes,* and *Haemophilus influenzae* [29,30,31,32,33,34,35].

However, there are many uncertainties regarding the impact of bacterial co-infections during the pandemic, especially in intensive care settings, that need to be evaluated for the sake of global health. The objective of this work was to determine the prevalence of bacterial co-infection in hospitalized patients with COVID-19 and to correlate the pathogenic bacteria that cause co-infection.

## 2. Results

### 2.1. Result of the Study Identification Process

A total of 542 records were identified from databases and manual searches. After remov ing duplicates (*n* = 29), 513 studies were screened by titles and abstracts, and 38 selected articles were eligible for full-text review. In 38 full-text studies, 10 were included in the review. Figure 1 provides details of the records excluded at each stage of the screening process.

### 2.2. Overall Co-Infections Result

The results of 10 articles were tabulated, resulting in a sample group of 1394 studies and 56% of the participants had bacterial co-infections. Among the species that stood out in this research, *Staphylococcus aureus* sensitive to methicillin was one of them with 27% (17/63) co-infections with 0.27 proportion (0.16–0.42; I^2^ = 65%). Another 16% (47/678) with 0.16 proportion (0.03–0.50; I2 = 97%) were co-infected with coagulase-negative *Staphylococcus* spp.

Another very important pathogen in terms of nosocomial/hospital infections and relevant in our research was *Klebsiella.* spp., with 21% (84/404) and 0.21 proportion (0.04–0.48; I2 = 96% and *p* < 0.001); *Klebsiella oxytoca* and *K. pneumoniae* were the second most expressive with 10% (9.9/99) co-infections and 0.10 proportion (0.03–0.30; I^2^ = 83%) and 3% (28/935), 0.03 (0.03–0.30; I^2^ = 97%), respectively.

The species *Enterobacter complex* proved to be very significant in this research with 42% (28/69) co-infections and 0.42 proportion (0.31–0.54); followed by 35% (48/138) infection caused by *Enterobacter* spp., with 0.35 proportion (0.25–0.46). Another species that proved to be important in the results of our study was *Escherichia coli* and extended-spectrum beta-lactamase-producing *Echerichia coli* (ESBL), both with 13% (10/80) co-infected patients in the respective 0.13 proportions (0.09–0.18; I2 = 63%; *p* < 0.02) and 13% (88/683) 0.13 (0.07–0.22; I2 = 15%; *p* < 0.28).

Other pathogens such as *Haemophilus influenza* and *Proteus mirabilis* showed 12% (39/330) co-infection and 0.12 proportions (0.03–0.40) and 11% (4/37) co-infection with 0.11 proportion (0.04–0.25). Given its biological importance, the species *Pseudomonas aeruginosa* appeared with 3% (30/1030) and 0.03 proportion (0.01–0.12) of co-infected patients in this study. Other species were also analyzed, as shown in Figure 2.

Some species did not show statistical significance for co-infection, such as *Sphingobacterium multivorum* and *Enterobacter cloacae*.

### 2.3. Co-Infections by Blood Flow, Urinary Tract, and Respiratory Tract Samples

The results for co-infections taking into account the biological materials from blood flow, urinary tract, and respiratory tract showed great relevance. This study describes the prevalence and characteristics of bacterial co-infection in patients with COVID-19 in the Intensive Care Unit (ICU) and hospitalized (not ICU).

#### 2.3.1. Blood Flow

The most common bacterial co-infections isolated from blood cultures include Coagulase-negative *Staphylococcus* with 44% (7/16) of patients with proportion 0.44 (0.22–0.68), *Enterobacter* spp. presented 14%/68 of patients with proportion 0.14 (0.03–0.44; I^2^ = 86%). Other data from other microorganisms can be seen in Figure 3.

#### 2.3.2. Respiratory Tract

The main results in relation to the respiratory tract were: *Streptococcus pneumoniae* presented in 57% (12/21) of patients and proportion 0.57 (0.36–0.76); *Klebsiella pneumoniae* with 33% (1/3) of patients with proportion 0.33 (0.04–0.85); *Pseudomonas aeruginosa* with 25% (16/64) of patients and proportion 0.25 (0.06–0.62; I^2^ = 42%). Other species were also important in this study: *Staphylocuccus aureus* accounted for 26% (18/71) of patients and proportion 0.26 (0.12–0.48; I^2^ = 51%); *Enterobacter* spp., with 21% (9/46) of patients and proportion 0.21 (0.02–0.75; I^2^ = 83%); *Kle* spp., for 9 (19%)/47 patients and proportion 0.19 (0.10–0.33; I^2^ = 56%); methicillin-resistant *Staphylococcus aureus* by 17% (8/51) and proportion 0.17 (0.02–0.64; I^2^ = 89%). Figure 3, shows the bacterial species and their proportions in relation to the respiratory tract.

#### 2.3.3. Urinary Tract

The pathogens found and relevant related to patients from urinary tract samples were: *Echerichia coli* presented in 37% (11/29) of patients with proportion 0.37 (0.20–0.59; I^2^ = 42%); *Klebsiella* spp. with 20% (2/10) of patients and proportion 0.20 (0.05–0.54); *Klebsiella pneumoniae* and *Enterococcus faecalis*, both with 17% (5/29) of patients and proportion 0.17 (0.07–0.35; I^2^ = 0%.); *Staphylococcus aureus* and *Proteus mirabilis*, both with 14% (1/7) of patients and proportion 0.14 (0.02–0.58); *Klebsiella variicola* and *Echerichia coli* (ESBL), both with 10% (1/10) of patients and proportion 0.10 (0.01–0.47). Figure 3 shows the bacterial species and their proportions in relation to the urinary tract.

## 3. Discussion

It is well known that seasonal viral respiratory tract infections are directly linked to an increased risk of bacterial co-infection. One of the main causes of mortality in previous influenza pandemics is bacterial infections. An assessment of bacterial respiratory tract co-infections in patients with COVID-19 admitted to the ICU also had secondary infections with Acinetobacter baumannii [36,37].

Chen et al. (2020) reported that this same microorganism infected by SARS-CoV-2 was significant among the species found in sputum material [38]. In another study focused on the same species, it was responsible for 20% of co-infections in ICU patients [39]. In our results, this bacterium was not important, but we know the importance of this bacterial species in the biological part regarding antimicrobial resistance; it is considered the first of the three main species related to antibiotic failure.

Several studies showed that bacterial co-infections in COVID-19 pneumonia are still emerging, but an association was made among the detection of bacterial pathogens with disease severity in COVID-19 patients. The most commonly identified co-infected bacteria include *Acinotobacter baumannii*, *Klebsiella pneumoniae*, *Legionella pneumophila,* and *Streptococcus pneumoniae* [40,41,42]. Such studies corroborate with our results, regarding the biological importance of these microorganisms in relation to antimicrobials, demonstrating the relevance of the study in relation to co-infections related to patients with COVID-19.

According to Martinez-Guerra and collaborators, 69 patients were studied, and it was found that the most frequent cause of healthcare-associated pneumonia in hospitalized patients was Enterobacteriaceae with 69.6% (48/69), followed by Gram-negative non-fermenting bacilli with 26.1% (18/69). The same authors studied 35 patients with bloodstream infections and found 40% (14/35) of samples positive for coagulase-negative *Staphylococcus* [43]. In our study, 16% (50/678) of patients were positive for coagulase-negative *Staphylococcus* for blood flow samples.

### 3.1. Blood Flow

A study was conducted with 92 adult patients, ICU admitted with acute respiratory failure, manifesting SARS-CoV-2 pneumonia; in total, (28%) 26 patients were co-infected with pathogenic bacteria on ICU admission; from these patients, 32 different bacteria were isolated from cultures and (31%) 10 of methicillin-sensitive *Staphylococcus*, 22% (7/32) Haemophilus influenza, and 19% (6/32) *Streptococcus pneumoniae* were found [44].

The prevalence of viral or bacterial co-infections in patients admitted to the ICU, for acute respiratory failure related to SARS-CoV-2 pneumonia is poorly studied [45,46]. One study reported 41% (7/17) co-infection among patients admitted to an ICU of a US hospital [47]. A cross-sectional study with 12.46% (5/43) positive cultures in blood samples and endotracheal aspirate samples were obtained and, *Klebsiella* spp. 25.59% (11/43), *Staphylococcus aureus* (MSSA) 20.93%, *Escherichia coli* 18.6% (8/43), *Staphylococcus aureus* (MRSA), 13.95% (6/43), *Enterobacter* spp. 11.63% (5/43), and *Pseudomonas aeruginosa* 9.30% (4/43) were isolated [48].

A study on central-line-associated bloodstream infection and catheter-associated urinary tract infection was carried out in 78 hospitals, considering the period prior to the COVID-19 pandemic and during the COVID-19 pandemic. It was found that the hospitals with COVID-19 patients had in one month 2.38 times higher co-infection, with <5% prevalence during the pandemic period (*p* = 0.004), when compared with a hospital with no COVID-19 patients. Coagulase-negative *Staphylococcus*-associated bloodstream infection had an increase of 130%/1000 days (*p* < 0.001) [49].

In our study, results similar to those of the described authors were found, with *Klebsiella* species 11%, proportion 0.11 (0.05–0.22); *Escherichia coli* 8% (6/71), proportion 0.08 (0.01–0.41); *Enterobacter* species 14% (10/68), proportion 0.14 (0.03–0.44); *Pseudomonas aeruginosa* 8% (6/71), proportion 0.08 (0.04–0.18); and 44% (7/16) for coagulase-negative *Staphylococcus aureus* with proportion 0.44 (0.22–0.69).

### 3.2. Urinary Tract

A retrospective observational study aimed to assess the mortality rate of 87 patients with COVID-19 from bacterial co-infections. The most common microorganisms were *Escherichia coli* 28.4% (27/87) and *Enterococcus faecalis* 26.3% (25/87) [50]. Our study corroborates this research, because we found co-infections for the same species *Escherichia coli* 37% (11/29), 0.37 [0.20–0.59]; *Enterococcus faecalis* 17% (5/29), 0.17 [0.07–0.35].

Another study carried out in Lahore (Pakistan), involving 130 pediatric ICU patients, isolated the pathogenic bacteria *Escherichia coli*, with 48% (63/130), and 28% (37/130) for *Klebsiella pneumoniae* [51]. A study with 242 patients positive for COVID-19 isolated 19% (46/242) bacterial co-infections. The most frequent was genitourinary representing 57% (138/242) of all co-infections and the most common organism was *Escherichia*
*coli,* 26% (63/242) [52]. Our study corroborates with the two last studies and confirms the presence of the same organisms in urinary tract co-infections with 20% (2/10), 0.20 [0.05–0.54] for *Klebsiella* species, and 37% (11/29), 0.37 [0.20–0. 59] for *Escherichia coli*.

### 3.3. Respiratory Tract

The prevalence for COVID-19 patients with respiratory co-infections is not well elucidated. Patients who trigger a more critical degree of the disease and need to be transferred to the Intensive Care Units (ICUs) end up needing the help of an intubation endotracheal tube after at least 48 h of mechanical ventilation and when undergoing this procedure, are susceptible to acquiring microbial pathogens. Due to high demand in the pandemic period, many hygiene protocols are not followed correctly, or are not performed, causing bacterial infections. Among people who are in the ICU with COVID-19, about 30% acquire secondary bacterial infections due to the procedure of intubation. Case studies show the presence of COVID-19 co-infections, including influenza in adults [53,54,55,56,57,58,59,60,61,62,63,64,65,66,67,68,69]. It has been proven that co-infections by Staphylococcus aureus or other bacteria during COVID-19 disease impair the innate and adaptive defenses of the host, temporarily compromising the physical and immunological barrier, causing secondary bacterial pneumonia, raising the severity and deaths in healthy people with pre-existing comorbidities [70].

In total, 836 patients have been studied with SARS-CoV-2 in respiratory culture samples and positivity was found in 13.3% (112/836) of samples and 112 positivity results, in 34.8% (39/112) was identified bacterial pathogens. The respiratory bacterial co-infections findings were *Staphylococcus aureus* 4% (4/24), as a community-acquired pathogen [50]. A retrospective cohort study analyzed 91.8% (236/243) of patients and they had bacterial co-infections with *Streptococcus pneumoniae* 59.5% (153/236), *Klebsiella pneumoniae* 55.6% (143/236), and *Haemophilus influenzae* 40.1% (103/236) [71].

Another retrospective cohort study analyzed 989 patients and showed 3.1% (31/989) and 3.74% (37/989) co-infections. Two of these co-infections were with different bacteria: *Streptococcus pneumoniae* in co-infection with *Moraxella*
*catarrhalis* and *Staphylococcus aureus* in co-infection with *Haemophilus influenzae* [72]. In our study, the most relevant species was *Streptococcus pneumoniae* with 57% (12/21) and proportion 0.57 (0.36–0.76) of co-infec tion and even though *Klebsiella pneumoniae* has biological relevance in relation to antimicrobial resistance, which has proved to be statistically irrelevant in this research.

In a retrospective analysis of 78 patients with COVID-19, 14.1% (11/78) were coinfected with respiratory pathogens that cause lung lesions, the most relevant being *Mycoplassma pneumoniae* 45.5% (5/11) [73]. In our study, the results corroborate this finding, only for *Legionella pneumophila* that presented 9% (1/11), a proportion of 0.09 (0.01–0.44) of co-infection that presented itself in a general way and not by blood flow or urinary and respiratory tracts.

A retrospective study analyzed 140 samples from the respiratory tract of ICU patients, positive for COVID-19, and detected 46% (23/50) positive for *Enterobacterales*, especially *Klebsiella* spp., 22% (11/50) [74]. It is important to emphasize that this bacterial genus is among the three main genera responsible for antimicrobial failure and in our study it was shown to be expressive in a general and specific way in ICU patients.

## 4. Materials and Methods

This systematic review was conducted based on the PRISMA (Figure 1) guidelines for reviews, to determine the proportions of co-infections in patients admitted to ICUs with COVID-19.

### 4.1. Study Identification

This research is characterized as a meta-analysis, with a search in the following databases: PubMed, Embase, and SciELO from December 2019 to December 2021. The search for references was performed using words found on the Health Sciences Descriptors website (DeCS/MeSH), considering: “Corona virus”, “SARS-CoV-2”, “COVID-19”, “bacterial infection”, “cross sectional”, and “co-infection”. The descriptors were associated with the Boolean operators AND, OR, and NOT. In PubMed, 479 references were found, in Embase 18, and in SciELO 44, totaling 541 publications; duplicate articles (*n* = 29) were later discarded and articles that did not meet the inclusion criteria (475), totaling 38 references used in this research. Out of 38 studies, 10 were selected for meta-analysis, following the Reporting Items for Systematic Reviews and Meta-Analyses (Figure 1) [75]. The evaluation process of the data extracted for meta-analysis was carried out by two statistical reviewers.

### 4.2. Inclusion and Exclusion Criteria

Inclusion criteria were: (1) retrospective, prospective, and cross-sectional studies of humans with laboratory-confirmed SARS-CoV-2 co-infection, clinical features, and (2) outcomes with ICU patients with co-infection. SARS-CoV-2 infections in blood culture, urinary tract, and respiratory tract.

Exclusion criteria were: (1) articles that did not report data on the number of patients with co-infection; (2) articles with less than 18 patients (defined as case series) and/or case reports; and (3) randomized controlled trials, reviews, editorials, animal studies, letters, and conference abstracts.

After identifying the studies that met all the inclusion and exclusion criteria, the full texts were reviewed for final inclusion in this research. Disagreements were independently reviewed by a researcher who did not participate in the screening.

### 4.3. Data Extraction

The authors independently extracted data from the included articles using a data collection form. Information was collected according to demographic variables: author and year of publication; country of study and healthcare environment (hospital ICU, non-ICU). This study considered and extracted data from cross-sectional studies, retrospective and prospective cohorts in order to assess the proportions of co-infections (Table 1).

### 4.4. Data Analysis

The primary outcome measure was the prevalence of co-infection among patients with COVID-19. Articles were classified by country to compare geographic prevalence. It was stratified by patients with co-infection and the co-infection identified with COVID-19 ICU patients was analyzed. The combination of data was performed using the proportion test, considering the generalized linear mixed model (GLMM), associated with logistic transformation (PLOGIT). Using this method, the proportion of patients with co-infection was determined and which was the most prevalent was established.

Heterogeneity was determined by the Higgins and Thompson test (I^2^). When the I^2^ presented a result equal to or greater than 50%, it was considered a randomized effect; on the other hand, when the I^2^ presented a result below 50%, the effect was considered fixed. For the significance of the results, the limit of 5% was considered. Statistical analyses were performed using RStudio^®^ 4.0.2 software [76] and Stata 16.0. [77].

## 5. Conclusions

We report that the highest rates of bacterial co-infection were for *Enterobacter complex* 42% (29/69), proportion 0.42 (0.31–0.54) and *Staphylococcus aureus* 27% (17/63), proportion 0.27 (0.16–0.42). Regarding ICU patients on admission, the co-infection positivity in blood flow was (44%), proportion 0.44 (0.22–0.42) for coagulase-negative *Staphylococcus*; in the respiratory tract, *Streptococcus pneumoniae* was found in 57% (12/21), proportion 0.57 (0.36–0.76) of the samples and in the urinary tract 37% (11/29), proportion 0.37 (0.20–0.59) of *Escherichia coli* were isolated. It is important to note that the species *Acinetobacter baumanii*, *Klebsiella pneumoniae,* and *Pseudomonas aeruginosa* are biologically relevant in this study, due to their history of resistance to antimicrobials and nosocomial/hospital infections.

## Figures and Tables

**Figure 1 antibiotics-11-00894-f001:**
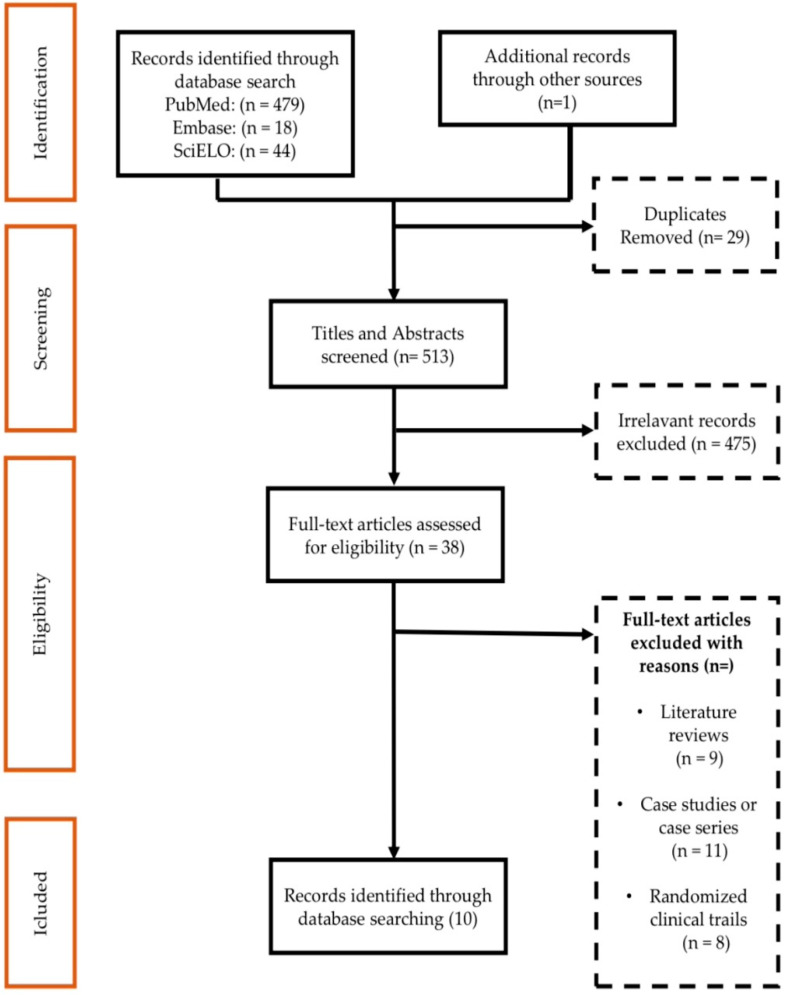
PRISMA schematic selection process of the included studies at each stage of the screening process.

**Figure 2 antibiotics-11-00894-f002:**
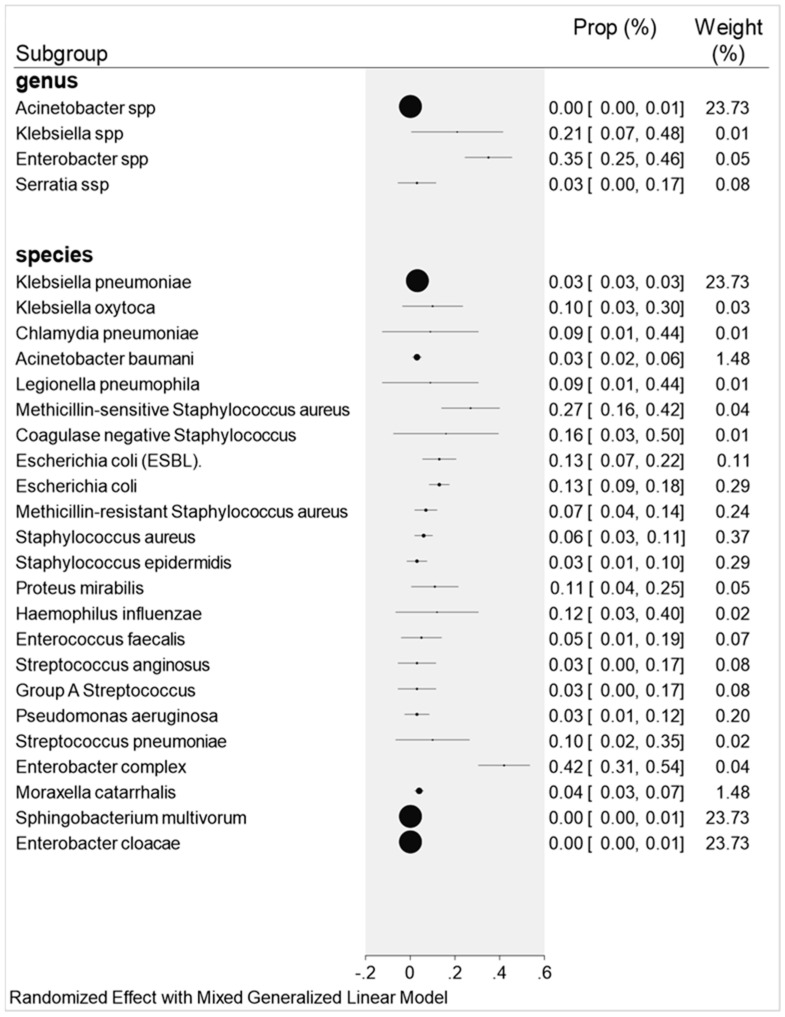
Forest plot graph: showing the subgroup (genus and bacterial species) and their respective proportions by co-infection.

**Figure 3 antibiotics-11-00894-f003:**
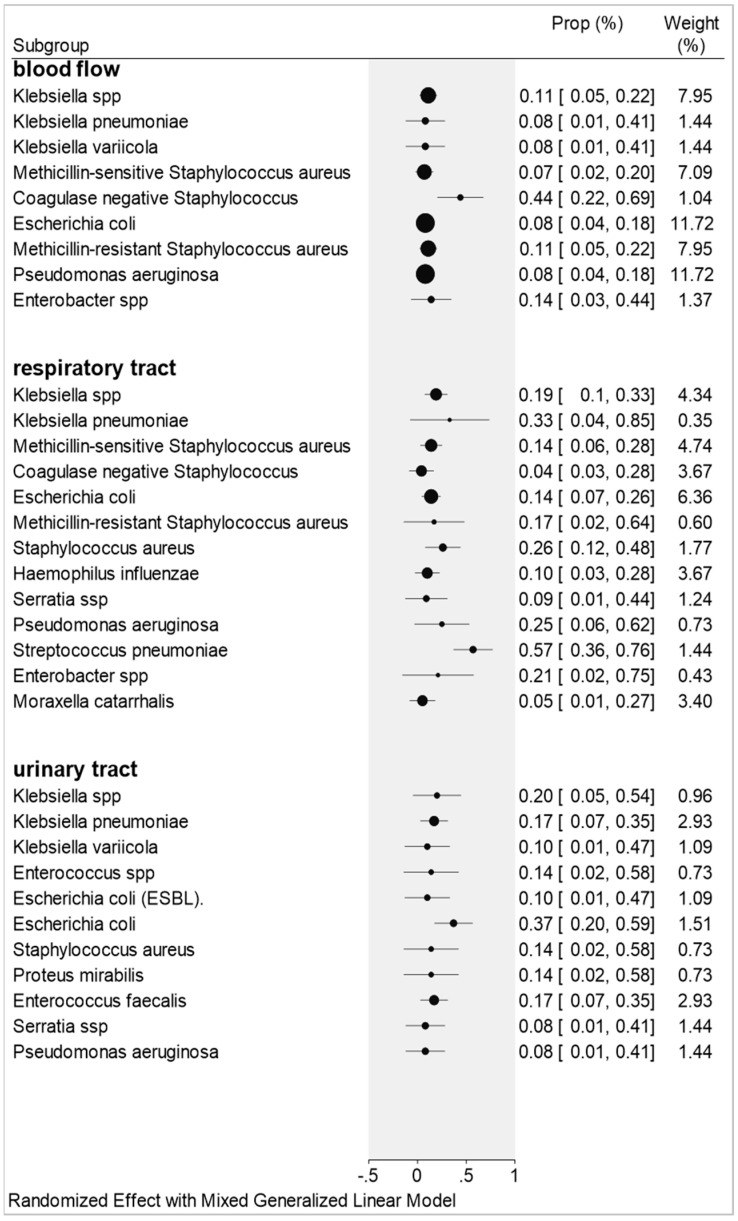
Bacterial species and their proportions in relation to blood flow, respiratory tract, and urinary tract.

**Table 1 antibiotics-11-00894-t001:** Research selected for meta-analysis.

Studies Selected for Meta-Analysis			
Author	Year	Countries	Type of Study
Wang et al. [40]	2021	London	Retrospective observational study
Martinez-Guerra et al. [45]	2021	Mexico	Prospective cohort study
Contou et al. [46]	2020	France	Retrospective study
Hughes et al. [49]	2020	London	Retrospective observational analysis
Mahmoudi [50]	2020	Iran	Cross-sectional study
Neto et al. [54]	2020	USA	Retrospective analysis
Zhu et al. [73]	2020	China	Retrospective study
Garcia-Vidal et al. [74]	2021	Spain	Retrospective cohort study
Man-Ling et al. [75]	2021	China	Retrospective analysis
Rothe et al. [26]	2021	Germany	Retrospective cohort study

## Data Availability

The data are contained in the article.

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
