# Peer review of "Bacterial Co-Infection in Patients with COVID-19 Hospitalized (ICU and Not ICU): Review and Meta-Analysis"

_antibiotics, 2022, doi:10.3390/antibiotics11070894_

Round 1

Reviewer 1 Report

The manuscript entitled: Bacterial Co-infection in Patients with COVID-19 Hospitalized in Intensive Treatment Units: Review and Meta-Analysis. It is a interesting and comprehensive 

meta-analysis of these retrospective study. It is relevant and worthy to read for clinicians. 

I suggest that the author should clarify the co-infection of respiratory tract into two group : 

initial admission ( < 2 days ) or nosocomial infections ( > 2 days ) .

Author Response

Answer letter

Revisor 1:

The manuscript entitled: Bacterial Co-infection in Patients with COVID-19 Hospitalized in Intensive Treatment Units: Review and Meta-Analysis. It is a interesting and comprehensive meta-analysis of these retrospective study.

It is relevant and worthy to read for clinicians.

I suggest that the author should clarify the co-infection of respiratory tract into two group:initial admission ( < 2 days ) or nosocomial infections ( > 2 days ) .

Authors: The prevalence for COVID-19 patients with respiratory co-infections is not well elucidated. Patients who trigger a more critical degree of the disease and need to be transferred to the Intensive Care Units (ICU), end up needing the help of intubation endotracheal tube after at least 48 hours of mechanical ventilation and when undergoing this procedure, are susceptible to acquiring microbial pathogens. Due to high demand in the pandemic period, many hygiene protocols are not done correctly, or are not performed, causing bacterial infections. Among people who are in the ICU with COVID-19, about 30% acquire secondary bacterial infections due to the procedure of intubation. Case studies show the presence of COVID-19 co-infections, including influenza in adults.

Reviewer 2 Report

Dear Author

The present study is a meta-analysis and literature review aiming at assessing the proportion of  bacterial co-infections in COVID-19 patients in the setting of ICUs. The authors also aim to “correlating the pathogenic bacteria causing co-infection.”

The study selected a restricted amount of papers by operating a standardized selection as described and calculated the proportion of patients presenting with secondary bacterial infections.

Randomized controlled trials and descriptive case reports as well as reviews were excluded, and case series involving less than 18 patients. Word selection was as follows:

The search for references was performed using words found on the Health Sciences Descriptors website (DeCS/MeSH), considering: “Corona virus”, “Covid”, “bacterial infection” “cross sectional”and “co-infection”.

The authors conclude that high rates of bacterial co-infections due to Enterobacter and S. aureus were reported, while A. baumannii, K. pneumoniae and P. aeruginosa infections pose a significant threat due to their resistance profile and hence represent a consistent burden in thenosocomial setting.

The revision comments are as follows:

1)     Explain search criteria: was SARS COV 2 also included? What population (only ICU or was also critically ill assessed? What time frame was the search performed in (eg 2020-2022?) What population (pediatric, adult, immunocompromised etc? Why were RCT excpluded?

2)     Language requires extensive revision. As it stands, the draft is of difficult understanding. Revision by a native English speaker or experiences scientific medical writer is strongly suggested for acceptance. The following sentences require revision:

·       35-38: not clear, maybe also irrelevant to the topic

·       49-51: unclear

·       55-57: unclear, explain why covid is susceptible to bacterial infections

·       64-65: there is a more recent report on the burden of AMR from Feb 2022

3)     Bacteria names should be uniformed to one format, either extended or abbreviated (eg E.coli or Escherichia coli )

4)     Explain the correlation described in the scope of the study “The objective of this work was determining the prevalence of bacterial co-infection in ICU patients with COVID-19, correlating the pathogenic bacteria causing co-infection.”

5)     The state of the art of co-infections and scope of the paper is poorly developed. The concept is confused, and should be clarified. As of today, there is confusing data concerning the impact and entity of co-infections, and possibly data from different methods of detection may determine the real burden. Please clarify

6)     Explain figures: why are dots with different sizes and what do they represent? How is the data calculated?

7)     Figure 2 states “forest plot” and is indicating Fig 2A, but does not correspond to the data Forest Plot graph: showing author and year, specific co-infection, total of co-infections 111 and etiologic agent.

8)     No data concerning the impact of resistance is reported, despite the initial introduction on the burden of resistance. Is there any correlation with outcome?

9)     Percentages and whole numbers should be expressed coherently by bracketing the whole number (eg line 123: 44% (44/???)).

10)  Explain sentence 113: A baumannii did not show clinical significance and the Fig 2 weight, along with the conclusions 153-158. Why is this data conflicting? Is this linked to secondary infections rather than co-infections?

11)  Data seems to be processed to analyze data according to geographical prevalence, but the data is neither reported nor discussed “Articles were classified by country to compare geographic prevalence. It was stratified by patients with co-infection and, analyzed the co-infection identified withCOVID-19 ICU patients.”

The article may be accepted following major revisions as requested above

Author Response

Answer letter

Revisor 2:

  1. a) Explain search criteria: was SARS COV 2 also included?

Authors: it was replaced

  1. b) What population (only ICU or was also critically ill assessed?

Authors: Hospitalized population

  1. c) What time frame was the search performed in (2019-2021)

Authors: it was replaced

  1. d) What population (pediatric, adult, immunocompromised etc?

Authors: All patients described in the studies were included.

  1. e) Why were RCT excluded?

Authors: Randomized controlled studies are research with methodologies that aim to have control over the population with reduction of bias through randomization, due to researchers' inference and not just observation, however, the present meta-analysis did not aim to estimate association , risk or difference among means, evaluated only the proportion of patients with COVID, coinfected by bacteria, so cross-sectional or hospitalized population articles were included, in this context, observational, without inference from the researchers.

2)     Language requires extensive revision.  As it stands, the draft is of difficult understanding. Revision by a native English speaker or experiences scientific medical writer is strongly suggested for acceptance. The following sentences require revision:

Authors: A native speaker made the revision

  • 35-38: not clear, maybe also irrelevant to the topic - it was replaced

Authors: “Respiratory viral and bacterial infections contribute substantially to the global burden of morbidity and mortality. Such simultaneous infections with the flu virus or bacteria that cause pneumonia, tend to make the patient's condition critical”

  • 49-51: nuclear - it was replaced

Authors: “Co-infection associated with viral pneumonia is the main cause of mortality and can considerably inhibit the host's immune system, which decreases the pharmacological response and makes the prognosis of the disease harmful.”

  • 55-57: unclear, explain why covid is susceptible to bacterial infections - it was replaced

Authors: “Sars-Cov-2 (COVID-19) leaves the body vulnerable to bacterial infections; however, this co-infection mechanism is not well understood but represents a threat to the respiratory epithelium favoring bacteremia [16-21]. A study carried out with ICU (Intensive Care Unit) patients in 88 countries showed that those patients who received at least one antibiotic during acute hospitalization, of these, more than half developed a secondary bacterial infection, requiring antibiotic therapy”

   64-65: there is a more recent report on the burden of AMR from Feb 2022 - it was replaced

Authors: “Relevant advances have been achieved and determined by the national AMR programme which is guided by the WHO Global Laboratory AMR Surveillance System (GLASS) in Uganda. Using the WHONET software, ARM data management was installed at the surveillance sites with trained personnel to guarantee the quality of the data. Six major pathogens that cause resistance-related deaths (Escherichia coli, followed by Staphylococcus aureus, Klebsiella pneumoniae, Streptococcus pneumoniae, Acinetobacter baumannii and Pseudomonas aeruginosa) were responsible for 929,000 deaths from ADR and 3.57 million (2.62–4.78) ADR-related deaths in 2019.

3)     Bacteria names should be uniformed to one format, either extended or abbreviated (eg E.coli or Escherichia coli ) - it was replaced

Authors: “We used the Echerichia coli format as an example for all species”

4)     Explain the correlation described in the scope of the study “The objective of this work was determining the prevalence of bacterial co-infection in ICU patients with COVID-19, correlating the pathogenic bacteria causing co-infection.”  

Authors: The objective of this study was to determine the prevalence of hospitalized patients with COVID-19. There are other viruses or bacteria that could be co-infected. In this case, co-infection of bacteria with Sar-Cov-2 virus was studied.

5)     The state of the art of co-infections and scope of the paper is poorly developed. The concept is confused, and should be clarified. As of today, there is confusing data concerning the impact and entity of co-infections, and possibly data from different methods of detection may determine the real burden. Please clarify

Authors: it was replaced

6)     Explain figures: why are dots with different sizes and what do they represent? How is the data calculated?

Authors: “They originated from segregated analyzes and at the end were joined in a single figure to reduce pollution. “

“Each point represents the final result of the segregated analyzes and the size of the points is related to the sample N used in each analysis.”

“The higher the sample N, the greater the weight of the result obtained, consequently, the lower the limits of the confidence intervals”

7)     Figure 2 states “forest plot” and is indicating Fig 2A, but does not correspond to the data Forest Plot graph: showing author and year, specific co-infection, total of co-infections 111 and etiologic agente.

Authors: it was replaced

“Figure 2A.  Forest Plot graph: showing the subgroup (genus and bacterial species) and their respective proportions by co-infection”

 8)  No data concerning the impact of resistance is reported, despite the initial introduction on the burden of resistance. Is there any correlation with outcome? Authors: “We removed the information despite bacterial resistance, since the focus in this work is co-infection related to covid-19”

9)     Percentages and whole numbers should be expressed coherently by bracketing the whole number (eg line 123: 44% (44/???)).

Authors: it was replaced. “All species mentioned in the article have been corrected”

10)  Explain sentence 113: A baumannii did not show clinical significance and the Fig 2 weight, along with the conclusions

Authors: it was replaced. “There was a misconception regarding the species Acinetobcter Baumanii that presented 3% of 262 patients. Acinetobacter spp, on the other hand, did not show statistical significance 0.00 (0.00-0.01), as well as Enterobacter clacae 0.00 (0.00-0.01) and Sphingobacterium multivorum with 0.00 (0.00-0.01). 0.01)”.

153-158. Why is this data conflicting? Is this linked to secondary infections rather than co-infections?

Authors: it was replaced. “Here only co-infection and not secondary infection”

11)  Data seems to be processed to analyze data according to geographical prevalence, but the data is neither reported nor discussed “Articles were classified by country to compare geographic prevalence.

Authors: It was stratified by patients with co-infection and, analyzed the co-infection identified withCOVID-19 hospitalized patients.”

Round 2

Reviewer 2 Report

Please include the following revisions: Title: hospitalized ICU and non-ICU Refer directly to figures in the text: line 111 “…as shown in figure 2.” Figures 2 and 3: please remove letters. There is only one figure and letters A and C are not needed.  Lines 123 and 180: Bloodstream not Blood flow. Please correct also eventually in text. 236: sentence begins with number, not correct. 267: correct English language of sentence. Proposal: “the present meta-analysis was conducted by searching the following…”